# From Historical Patches to Repair Plans: Outcome-Conditioned Reasoning for Repository-Level Program Repair

Chenglin Li [1]   Yisen Xu [1]   Zehao Wang [1]   Shin Hwei Tan [1]   Tse-Hsun (Peter) Chen [1]

## Abstract

Repository-level automated program repair (APR) requires long-horizon reasoning over interdependent decisions. However, most LLM-based approaches reconstruct repair reasoning independently for each issue, failing to reuse successful patterns from prior repairs, even though real-world repositories contain many related issues with shared structure or constraints. Existing methods typically rely on forward exploration, which operates under outcome uncertainty, incurs substantial inference-time overhead, and can drift from the final correct patch. We propose Conditional Reasoning Distillation (*ConRAD*), which leverages in-repository resolved issues by reconstructing repair reasoning backward from verified patches and distilling outcome-consistent, stage-wise repair reasoning plans. Injected at inference time, these plans guide fault localization and patch generation, replacing open-ended exploration with constrained inference without fine-tuning or search. On SWE-Bench Lite, *ConRAD* improves Pass@1 by 10.4% (GPT-4o), 8.6% (DeepSeek-V3), and 10.3% (GPT-5), demonstrating a scalable inference-time alternative to forward exploration for long-horizon APR. The code is publicly available at https://github.com/Reasoning4Code/ConRAD.

## 1. Introduction

Repository-level automated program repair (APR) involves long sequences of reasoning steps (Yang et al., 2025a; Gao et al., 2025; Ma et al., 2024). To resolve a real-world issue, a large-language model (LLM) must interpret the issue description, identify relevant files and methods, understand project-specific constraints, and generate a correct patch (Mu et al., 2025; Yang et al., 2025b). These stages are tightly coupled: errors made early—such as incorrect fault localization or misinterpreting context—often propagate downstream, leading to cascading failures. Consequently, repository-level APR is fragile, as success depends on making consistent decisions across many interdependent steps.

Despite this complexity, most existing LLM-based APR research treats each issue as an independent reasoning problem at inference time (Xia et al., 2025; Yang et al., 2024; Wang et al., 2024; Zhang et al., 2024b). For every new issue, the model reconstructs a complete reasoning trajectory from scratch, without retaining or leveraging successful reasoning patterns from prior repairs. This stands in contrast to real-world software development, where repositories often contain related issues that share code structure, failure modes, or repair constraints. Leveraging reasoning from previously resolved issues, therefore, offers an opportunity to guide intermediate decisions and reduce the likelihood of inconsistent or misleading reasoning paths.

However, reusing repair reasoning across issues is challenging under realistic conditions. Many real-world issues lack reproducible tests, runnable environments, or reliable execution oracles (Qi et al., 2015; Wang et al., 2025b), making it difficult to generate or validate detailed repair reasoning during the repair process itself. Moreover, even when successful repairs exist, identifying which prior issues provide reasoning that transfers meaningfully to a new problem remains non-trivial (Liu et al., 2023). Superficial similarities between bugs do not guarantee alignment in underlying repair logic, and indiscriminate reuse risks introducing misleading assumptions (Gao et al., 2023; Li et al., 2025c).

To cope with the difficulty of long-horizon reasoning under limited supervision, several recent approaches rely on forward reasoning generation or search-based strategies, such as iterative refinement (Pan et al., 2024; Ma et al., 2024; 2025c) or Monte Carlo Tree Search (MCTS) (Wang et al., 2025c). These methods explore multiple candidate reasoning trajectories for a given issue, using heuristics, model confidence, or execution signals to guide the search. While forward exploration can help recover from early mistakes, it still constructs reasoning under outcome uncertainty and typically focuses on solving the current issue rather than

---

[1]Concordia University, Montreal, Canada. Correspondence to: Chenglin Li <chenglin.li@mail.concordia.ca>.

*Proceedings of the 43rd International Conference on Machine Learning*, Seoul, South Korea. PMLR 306, 2026. Copyright 2026 by the author(s).

producing reusable reasoning.

Developers' provided fixes define a concrete outcome that repair reasoning can condition on. Given a resolved issue and its ground truth patch provided by developers, reasoning can be reconstructed by working backward from the known outcome, constraining intermediate decisions to remain compatible with the final fix. This outcome-conditioned reconstruction removes the need for trial-and-error exploration and yields reasoning plans that are globally consistent with the ground-truth patch.

Building on this idea, we propose Conditional Reasoning Distillation (*ConRAD*), a framework that distills such outcome-consistent, stage-wise repair reasoning from historically resolved issues and injects it at inference time to guide new repair tasks. By replacing open-ended exploration with constrained inference, *ConRAD* converts individual historical repairs into structured, reusable procedural reasoning, even when execution environments or reliable test oracles are unavailable. To ensure reliable transfer, *ConRAD* further incorporates a conservative filter mechanism, Exemplar Guardian, that discards historical issues likely to provide misleading guidance.

We evaluate *ConRAD* on SWE-Bench Lite across three LLM backbones using Agentless (Xia et al., 2025) as the execution scaffold. *ConRAD* consistently improves end-to-end repair success, raising Pass@1 from 27.3% to 37.7% with GPT-4o, from 27.0% to 35.6% with DeepSeek-V3, and from 29.7% to 40.0% with GPT-5. Beyond that, *ConRAD* also improves fault localization accuracy, indicating that the distilled plans guide earlier decisions in the repair pipeline. Compared to an MCTS-based forward-search baseline, *ConRAD* achieves +13.0% in Pass@1 in a controlled comparison, using 9.71 times fewer LLM calls and 2.47 times fewer tokens. These results indicate that *ConRAD* provides a scalable inference-time alternative to forward exploration for long-horizon reasoning APR tasks. The key contributions of this work are as follows:

- We propose *ConRAD*, a framework that distills reusable, outcome-conditioned repair reasoning plans from historically verified fixes via Outcome-Conditioned Reasoning Distillation, enabling inference-time guidance without fine-tuning or large-scale iterative search.

- We evaluate *ConRAD* on SWE-bench Lite (Jimenez et al., 2023) across three LLM backbones and show consistent improvements over strong baselines under identical model settings, including higher end-to-end success and improved fault localization.

- Ours controlled comparison analysis shows that *ConRAD* achieves gains with substantially lower inference-

time overhead than forward-search refinement, highlighting backward distillation as a compute-efficient alternative to forward exploration for repository-level APR.

**Conflict of Interest Disclosure** The authors declare no financial conflicts of interest related to this work.

## 2. Background and Related Works

### 2.1. Large Language Models for Automated Program Repair

Early work framed automated program repair (APR) as localized patch synthesis (Chen, 2021; Fried et al., 2022) or iterative conversational refinement (Wang et al., 2023; Xia & Zhang, 2024). However, these approaches often struggle with repository-level dependencies.

To address repository-level dependencies, agentic systems, such as SWE-agent (Yang et al., 2024) and Open-Hands (Wang et al., 2024), equip LLMs with tool-use capabilities for iterative code navigation and modifications. Other works further incorporate structural analysis, test generation or task decomposition to guide context selection (Zhang et al., 2024b; Arora et al., 2024; Ahmed et al., 2025). Agentless (Xia et al., 2025) shows that a carefully designed agent workflow can achieve competitive performance without extensive online exploration, highlighting the importance of structuring the repair process. PatchPilot (Li et al., 2025a) proposed a rule-based planning workflow with a refinement step to improve patch quality. Some studies treat repository-level repair as a search-based repair agent (Ma et al., 2024; 2025b), often incorporating search-style refinement (e.g., MCTS) to explore multiple candidates (Hu et al., 2025; Antoniades et al., 2024). To further leverage historical data, retrieval-based methods (Liu et al., 2025; Zhang et al., 2025) and memory mechanisms (Guo et al., 2025; Mu et al., 2025) attempt to reuse patterns or heuristics from past issues.

Unlike prior methods that rely on narrow pattern transfer or lack cross-issue reuse, ConRAD distills general, outcome-conditioned reasoning plans from verified fixes to provide direct inference-time guidance across diverse issues.

### 2.2. Reasoning Elicitation in Program Repair

Prior work shows that explicit reasoning strategies are critical for complex program repair tasks (Wei et al., 2022; Li et al., 2025b). Chain-of-thought (CoT) prompting and self-correction approaches allow models to articulate intermediate reasoning or iteratively refine patches using execution feedback (Yin et al., 2024; Pan et al., 2024; Ma et al., 2025a). A related line of work extracts CoT data from historically resolved issues and uses it as supervision

for training or fine-tuning repair models. To ensure CoT quality, existing studies typically apply validation or filtering (Ma et al., 2025c; Pan et al., 2024; Xie et al., 2025; Wei et al., 2025). Execution-based validation can further improve CoT correctness, but often incurs substantial per-instance environment cost and can be difficult to scale (Pan et al., 2024). Beyond training-time supervision, some studies explore multiple reasoning trajectories at inference time via sampling or searching. In particular, MCTS-style methods integrate structured planning and tree search to expand and select promising repair paths (Wang et al., 2025c; Antoniades et al., 2024). Yet these approaches mostly reason *forward* under outcome uncertainty, incurring high inference cost without guaranteeing global consistency, and still require validation against test-passing fixes (Huang et al., 2023).

While retrospective reasoning reconstruction has shown promise in open-ended tasks (Wang et al., 2025a), repository-level APR requires code-grounded, procedural *plans* rather than free-form rationales. *ConRAD* distills such structured guidance from verified fixes, enabling efficient inference-time repair without the overhead of extensive search.

## 3. Approach

We introduce *ConRAD*, a framework for *outcome-conditional reasoning transfer* that enables LLM test-time scaling by distilling reusable repair guidance from historically resolved issues with ground-truth fixes.

Figure 1 shows the overall workflow of *ConRAD*, which consists of three stages: **(1) Repository-Level Exemplar Mining** retrieves candidate historical bug issues from the same repository by combining textual similarity with semantic alignment, ensuring consistency in code structure and project-specific conventions; **(2) Exemplar Guardian** acts as a conservative compatibility filter, evaluating whether a retrieved exemplar is likely to provide transferable repair guidance for the target bug and discarding exemplars deemed misleading; **(3) Outcome-Conditional Reasoning Distillation** derives a structured, conditional repair plan from the ground-truth fix of a validated exemplar, abstracting procedural knowledge that can guide inference-time reasoning during the repair of new bugs.

### 3.1. Stage 1: Repository-Level Exemplar Mining

The first stage of *ConRAD* identifies a historically resolved bug issue from the *same repository* whose resolution may provide transferable guidance for repairing a new target bug. Instead of relying solely on textual similarity, this stage aims to identify *candidate exemplars* aligned with the target issue in terms of problem intent and code context.

*ConRAD* constructs a repository-specific historical bug dataset comprising resolved bugs, their corresponding ground-truth fixes, and associated timestamps. Restricting retrieval to the same repository helps preserve consistency in naming conventions, architectural patterns, and dependency usage (Jiang et al., 2018), which are often critical for meaningful transfer in program repair and difficult to capture through text similarity alone. To prevent temporal leakage, only bug issues resolved strictly before the target issue's creation time are considered eligible exemplars.

Given a target bug issue, *ConRAD* performs a two-phase retrieval procedure that combines lightweight textual filtering with semantic alignment. First, it retrieves a small set of candidate historical issues based on textual similarity. Specifically, *ConRAD* encodes issue summaries using *All-MiniLM-L6-v2*, a pretrained sentence embedding model from Sentence Transformers (Reimers & Gurevych, 2019), and computes cosine similarity between the target issue and all eligible historical issues within the repository. *ConRAD* retains the top five candidates as textually similar exemplars, balancing recall with the cost and noise of the subsequent LLM-based semantic filtering. Second, *ConRAD* refines the candidate set via a semantic compatibility assessment that selects the most aligned historical issue with respect to the target's failure intent and repair context. Following prior work on *LLM-as-a-judge* (Zheng et al., 2023), the assessment applies a structured comparison rubric to assess whether each candidate issue's underlying problem intent is semantically compatible with that of the target issue. The rubric specifies multiple comparison dimensions, including failure symptoms, affected components, and implicit repair intent. The full prompt is provided in Appendix C. Based on this assessment, the LLM selects a single historical issue judged to be most semantically aligned with the target bug.

### 3.2. Stage 2: Exemplar Guardian

As *ConRAD* distills and transfers repair reasoning plan from the retrieved exemplar to the target issue, it is crucial that the exemplar corresponds to a *compatible debugging scenario*. In practice, example-guided repair is highly sensitive to analogy quality: two issues may appear similar at the surface level yet differ in their underlying causal mechanisms or repair constraints. Conditioning on such mismatched exemplars can bias the model toward irrelevant analogies and induce *negative transfer*, resulting in spurious reasoning steps and misleading fixes (Nguyen & Wong, 2023; Ye et al., 2023).

To mitigate this risk, *ConRAD* employs an **Exemplar Guardian**, a LLM-based compatibility filter that screens the retrieved exemplar *before* any reasoning distillation occurs. The Guardian's objective is not to re-rank candidates by generic similarity, but to estimate whether the exemplar

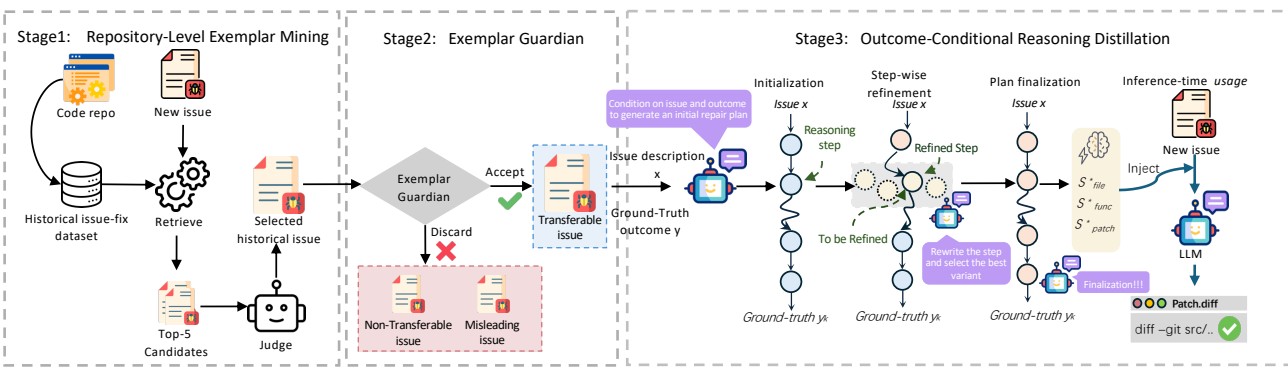

*Figure 1. ConRAD* overview: Stage 1 retrieves in-repository exemplars and selects one via an LLM-based judge. Stage 2 filters candidates into Transferable, Non-transferable, Misleading. Stage 3 distills outcome-conditioned plans $S^*_{file}, S^*_{func}, S^*_{patch}$ from the exemplar's ground-truth fix and injects them as in-context guidance during inference.

can provide *transferable diagnostic or procedural insight* for the target issue, given differences in failure context and repair constraints. Importantly, the Guardian is intentionally designed to be conservative: when uncertainty exists, it prefers discarding an exemplar to the risk of transferring misleading repair guidance.

To ensure reliable judgments, the Guardian enforces a structured, criterion-based evaluation rubric inspired by meta-prompting principles (Zhang et al., 2023). An LLM-based evaluator applies this rubric to produce verification decisions. Specifically, it assesses the compatibility between the target and the exemplar along five dimensions (Tan et al., 2024; Giamattei et al., 2025): *Root-Cause Similarity*, *Causal-Chain Transferability*, *Fix-Strategy Applicability*, *Contextual Alignment*, and *Debugging-Technique Relevance*. The details of the rubrics are provided in Appendix D.

The Guardian produces (i) a brief justification, (ii) a confidence estimate, and (iii) a continuous *transferability score* $s \in [-1, 1]$, where larger values indicate higher expected transferability and negative values indicate likely misalignment. The discrete decision is derived by a rubric-based assessment over the five criteria above: the exemplar is labeled **Transferable** when the criteria jointly support actionable and context-aligned diagnostic/procedural guidance; it is labeled **Misleading** when it appears superficially similar yet is likely to induce negative transfer (e.g., steering localization or edits toward an incorrect component or fix pattern); otherwise it is labeled **Non-transferable**. Only exemplars labeled **Transferable** are *accepted* and proceed to the distillation stage; **Non-transferable** and **Misleading** exemplars are *discarded*, in which case ConRAD falls back to the unmodified repair scaffold without injecting exemplar guidance. The score $s$ is recorded as an auxiliary signal for analysis and downstream control.

To stabilize decisions, we incorporate a one-step self-reflection (Shinn et al., 2023), in which the Guardian re-

evaluates its initial judgment based on the original score, confidence, and rationale. The final revised decision is passed to the next step.

### 3.3. Stage 3: Outcome-Conditional Reasoning Distillation

In the final stage, *ConRAD* applies *Backward Reasoning Distillation* (BRD) to derive a structured, transferable repair plan from a validated exemplar. Rather than reusing the exemplar's fix or its surface-level rationale, BRD reconstructs a sequence of reasoning steps that are *constrained by the exemplar's ground-truth outcome*. These steps form a conditional repair plan that can be reused to guide the repair of a new target issue. The distilled plans are injected at inference time as in-context guidance, enabling transfer without parameter updates or retraining.

BRD is motivated by a key asymmetry in reasoning-based inference. Traditional chain-of-thought and planning approaches generate *forward* reasoning, sampling intermediate steps without knowing whether they will lead to a correct solution, as in forward search methods such as Monte Carlo Tree Search (Hao et al., 2023; Zhang et al., 2024a). In contrast, BRD conditions reasoning generation on a *known-correct terminal state*. By explicitly conditioning on both the exemplar's issue description and its verified fix, BRD biases the model toward reasoning trajectories that are globally consistent with a valid repair, rather than merely locally plausible. Importantly, the objective is not to recover the original developer's reasoning, but to construct a *retrospectively valid and reusable repair plan* whose purpose is transferability rather than historical fidelity.

Formally, during reasoning reconstruction, BRD exploits the autoregressive nature of LLMs, where each generated token is conditioned on all preceding contexts. Given structured conditioning information—including the exemplar's issue description $x$, the ground-truth output $y_k$ for a given re-

pair sub-task, and explicit bug-fixing guidance (i.e., a fixed natural-language instruction that specifies the sub-task goal, constraints, and the desired plan format)—the model performs standard autoregressive decoding. Concretely, at each step, the model produces the next token $t_i$ conditioned on the structured information and previously generated tokens $t_{<i}$:

$$P_\theta(t_i \mid t_{<i}) \ \rightarrow \ P_\theta(t_i \mid t_{<i}, x, y_k, \text{guidance}),$$

This reformulation replaces unconstrained forward reasoning $P_\theta(t_i \mid t_{<i})$ with a conditional next-token prediction objective $P_\theta(t_i \mid t_{<i}, x, y_k, \text{guidance})$ that imposes a global outcome constraint. Conditioning on $y_k$ narrows the space of plausible continuations to those compatible with a verified fix, thereby reducing vague, speculative, or internally inconsistent reasoning steps (Wei et al., 2022; Li et al., 2023). Consequently, the reconstructed reasoning is causally consistent and explicitly aligned with a known-correct repair.

**Problem setup.** For each exemplar, we consider three repair sub-tasks $k \in \{\text{file}, \text{function}, \text{patch}\}$, corresponding to file-level fault localization, function-level fault localization, and patch synthesis. Given the exemplar issue description $x$ and the ground-truth outcome $y_k$ for sub-task $k$, BRD reconstructs a natural-language reasoning plan

$$S_k = [s_{k,1}, \ldots, s_{k,n_k}],$$

where each step $s_{k,i}$ represents a concrete diagnostic or repair decision that bridges the issue description and the verified outcome. Each step is generated in a separate forward pass under a structured prompt that enforces concreteness, causal grounding, and consistency with both the surrounding steps and the final fix. This step-wise generation strategy stabilizes conditioning, avoids uncontrolled reasoning drift, and yields a modular reasoning plan that can be selectively reused during inference on new target issues.

**Step 1: Initial plan construction.** BRD first generates a coarse reasoning plan for each sub-task by prompting the LLM with (i) the exemplar issue description $x$, (ii) the corresponding ground-truth outcome $y_k$, and (iii) a structured meta-prompt that specifies the desired properties of a repair plan (e.g., concreteness, causal grounding, and consistency with the fix). The three sub-tasks are generated sequentially: file-level reasoning $S_{\text{file}}$ is produced first, followed by function-level reasoning $S_{\text{func}}$ conditioned on the file-level plan, and finally patch-level reasoning $S_{\text{patch}}$ conditioned on both. This yields an initial plan:

$$S_{\text{init}} = \{S_{\text{file}}, S_{\text{func}}, S_{\text{patch}}\},$$

which reflects a complete but potentially coarse repair reasoning plan consistent with the exemplar's fix, from a higher-level localization decision to final patch generation.

**Step 2: Step-wise refinement via local re-conditioning.** The initial plan may contain vague or weakly grounded steps. We represent each plan $S_k$ as an ordered sequence of steps $[s_{k,1}, \ldots, s_{k,n_k}]$. BRD therefore refines the plan one step at a time. For a given step $s_{k,i}$, we define its surrounding context as the remaining steps

$$\bar{S}_{k,i} = [s_{k,1}, \ldots, s_{k,i-1}, s_{k,i+1}, \ldots, s_{k,n_k}],$$

with their original order preserved. The model is then re-prompted with (i) the exemplar issue $x$, (ii) the surrounding plan context $\bar{S}_{k,i}$, (iii) the step marked for revision, and (iv) the full set of exemplar ground-truth outcomes $\mathcal{Y} = \{y_{\text{file}}, y_{\text{func}}, y_{\text{patch}}\}$.

Including $\mathcal{Y}$ imposes a global constraint during refinement: each revised step must remain compatible not only with its local context, but also with the final verified patch. This encourages each step to function as a coherent link in a globally consistent repair plan, rather than an isolated explanation.

For each step $s_{k,i}$, *ConRAD* performs *refinement* by generating three candidate rewrites and selecting the best option that preserves consistency with both the surrounding steps and the outcome constraint. Specifically, the selection is performed via a comparison that jointly ranks the original step $s_{k,i}$ and the three rewrite candidates, and selects the top-ranked option as $s^*_{k,i}$. The comparison prioritizes (1) consistency with the surrounding context $\bar{S}_{k,i}$, (2) compatibility with the verified outcomes $\mathcal{Y}$, and (3) specificity and actionability, while penalizing spurious or distracting detours that could mislead downstream repair. We perform a single refinement pass over all steps in $S_k$ (Prompt templates are provided in Appendix E.2).

**Step 3: Plan finalization and inference-time usage.** After all steps have been refined, BRD outputs a finalized repair plan

$$S^*_k = [s^*_{k,1}, \ldots, s^*_{k,n_k}],$$

for each sub-task $k$. Collectively, these plans form

$$S^* = \{S^*_{\text{file}}, S^*_{\text{func}}, S^*_{\text{patch}}\}.$$

The distilled plans in $S^*$ are inserted into the prompt when repairing a new target issue, providing the model with a structured demonstration of how a similar bug was diagnosed and resolved. BRD operates at inference time and requires no fine-tuning, making it compatible with standard autoregressive LLMs.

**Compatibility with other agentic APR scaffolds.** *ConRAD* can be integrated into other agentic frameworks. In this work, we evaluate reasoning distillation using a fixed

scaffold to isolate its contribution. However, as with Agentless (Xia et al., 2025), frameworks like SWE-agent (or mini-SWE-agent) (Yang et al., 2024) conceptually decompose repair into localization, repair, and testing stages, making our reasoning distillation approach applicable in these settings. While SWE-agent may revisit these stages through an exploratory loop under uncertainty, *ConRAD*'s reasoning distillation is conditioned on verified fixes and therefore does not rely on fallback exploration. As a result, the distilled reasoning plan can be integrated in the prompt to provide guidance on the agent's decisions across iterations (e.g., where to search, what evidence to look for, and what properties a valid fix should satisfy), without changing the agent's design and execution logic (more details in Appendix G).

## 4. Experiment Setup

In this section, we describe the evaluation benchmark, metrics, base models, and baselines. The implementation details can be found in Appendix A.

### 4.1. Benchmark

We conduct our experiments on SWE-Bench Lite (Jimenez et al., 2023), a widely used, cost-efficient, and representative subset of the original SWE-Bench benchmark. SWE-Bench Lite contains 300 issues collected from real-world projects. Each issue includes a ground-truth patch and a set of tests for automatically verifying the correctness of newly generated patches. Prior work has shown that SWE-Bench Lite preserves the relative difficulty and diversity of the full benchmark while substantially reducing evaluation cost, making it suitable for controlled comparative studies (Yang et al., 2024; Xia et al., 2025).

### 4.2. Evaluation Metrics

Following the previous studies (Xia et al., 2025), we evaluate the efficacy of *ConRAD* using three quantitative metrics: (1) **Pass@1** measures the percentage of issues for which the single generated patch (one attempt per issue) passes the complete test suite. (2) **File Localization Accuracy (%File)** and (3) **Function Localization Accuracy (%Func)** evaluate whether the locations modified by the model-generated patch cover the ground-truth edit locations at the file and function granularity, respectively. Let $F^\star$ be the set of files or functions modified in the developer patch and $\hat{F}$ be the corresponding set of locations modified by the model. A prediction is considered *correct* if $F^\star \subseteq \hat{F}$. We report the accuracy separately for files and functions. All metrics are computed over the full evaluation set using a single generated patch per issue.

### 4.3. Framework and Base Models

We use Agentless (Xia et al., 2025) as the execution scaffold for *ConRAD*. Agentless provides a lightweight, fixed-stage workflow for localization and patch generation, enabling a controlled and reproducible evaluation of reasoning distillation. Importantly, *ConRAD* does not rely on Agentless-specific mechanisms and is compatible with other repair pipelines. We evaluate *ConRAD* using three base LLMs: GPT-4o (`gpt-4o-2024-05-13`) (OpenAI, 2024), GPT-5 (`gpt-5-2025-08-07`) (OpenAI, 2025) and DeepSeek-V3 (`deepseek-v3-2025-03-24`) (Liu et al., 2024). GPT-4o is the primary backbone because of its strong performance and widespread adoption in recent software engineering agents, enabling direct comparison with prior work. To assess cross-model generalizability, we additionally evaluate *ConRAD* on GPT-5 and DeepSeek-V3. We use the same base model across all the stages of the pipeline.

### 4.4. Baselines

We compare *ConRAD* against seven highest-performing baselines (at the time of submission) on the SWE-Bench Lite leaderboard that use the same base LLM (GPT-4o), ensuring fair comparison. Specifically, we include Agentless (Xia et al., 2025) (the execution scaffold) and representative state-of-the-art methods including SWE-Agent (Yang et al., 2024), OpenHands (Wang et al., 2024), AutoCodeRover (Zhang et al., 2024b), MoatlessTools (Moatless Tool Team, 2024), MASAI (Arora et al., 2024), and SWE-Search (Antoniades et al., 2024).

## 5. Results

### 5.1. Localization and Resolution Results

**Localization Accuracy and Pass@1 of *ConRAD*.** Table 1 reports the performance of *ConRAD* and baseline methods on SWE-Bench Lite. For GPT-4o, we use baseline results reported in prior studies (Yang et al., 2024; Wang et al., 2024; Zhang et al., 2024b; Xia et al., 2025; Moatless Tool Team, 2024; Arora et al., 2024; Antoniades et al., 2024), which were evaluated under the same base model. For GPT-5 and DeepSeek-V3, where results are unavailable for most baselines, we rerun Agentless with the corresponding base model for comparison.

Across all three base LLMs, *ConRAD* delivers consistently competitive results. Notably, *ConRAD* yields statistically significant improvements over Agentless across all evaluated models (McNemar's test, $p < 0.001$). Specifically, with GPT-4o, *ConRAD* achieves a 37.7% Pass@1, outperforming the strongest competitor (SWE-Search) by 6.7% and its direct scaffold (Agentless) by 10.4%. *ConRAD* also shows improved fault localization accuracy, where it achieves 59.3% function level accuracy, surpassing Agent-

*Table 1.* Localization accuracy (%File, %Func) and Pass@1 for *ConRAD* and baselines across three base LLMs. '-' indicates that the information is not available.

| Approach | %File | %Func | Pass@1 (%) |
|---|---|---|---|
| *GPT-4o* | | | |
| SWE-Agent | 58.3 % | 42.3 % | 18.3% (55/300) |
| OpenHands | - | - | 22.0% (66/300) |
| AutoCodeRover | 62.3% | 42.3% | 22.7% (68/300) |
| MoatlessTools | 73.3 % | 52.0% | 25.0% (75/300) |
| Agentless | 68.7% | 51.0% | 27.3% (82/300) |
| MASAI | 75.0% | 56.3% | 28.0% (84/300) |
| AutoCodeRover-v2 | 69.3% | 52.3% | 30.6% (92/300) |
| SWE-Search | - | - | 31.0% (93/300) |
| *ConRAD* | **77.0%** ↑ | **59.3%** ↑ | **37.7%** (113/300) ↑ |
| *GPT-5* | | | |
| Agentless | 85.0% | 69.0% | 29.7% (89/300) |
| *ConRAD* | **86.3%** ↑ | **69.7%** ↑ | **40.0%** (120/300) ↑ |
| *Deepseek-v3* | | | |
| Agentless | 77.0% | 63.0% | 27.0% (81/300) |
| MoatlessTools | - | - | 30.7% (92/300) |
| *ConRAD* | **80.3%** ↑ | **66.0%** ↑ | **35.6%** (107/300) ↑ |

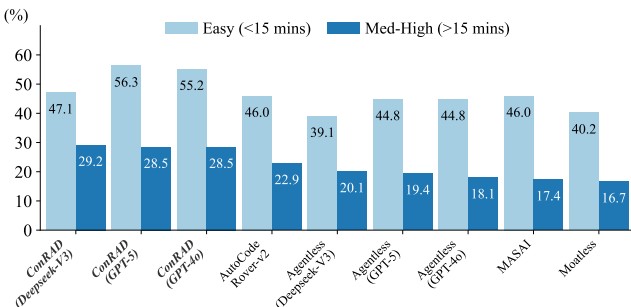

*Figure 2.* Comparison of Pass@1 results on tasks grouped by different difficulty levels.

less by 8.3%.

We observe similar trends when using GPT-5 and DeepSeek-V3, indicating the benefits of *ConRAD* generalize across different base models. On GPT-5, *ConRAD* improves Pass@1 from 29.7% to 40.0% over its Agentless scaffold while maintaining comparable localization accuracy. On DeepSeek-V3, *ConRAD* attains a 35.6% Pass@1, improving over Agentless by 8.6% and outperforming MoatlessTools by 4.9%. Although we observe less improvement in fault localization accuracy, *ConRAD* consistently generates more correct patches after accurate fault localization.

**Pass@1 on Different Difficulty Levels.** We compare *ConRAD* and baselines across difficulty levels on the subset of SWE-bench Lite issues with available difficulty annotations (277 out of 300), provided by OpenAI (OpenAI, 2024). These labels estimate the time required by experienced software engineers to produce a correct patch for

each issue. Due to the limited number of higher-difficulty cases, we group the annotations into two categories, *Easy* and *Medium-High*.

We conduct a difficulty-stratified analysis by calculating the Pass@1 for each issue within each difficulty group. As the analysis requires detailed results for the calculation, we only baselines where we have access to the per-issue results (e.g., SWE-Search is excluded as the per-issue data is unavailable). Figure 2 shows the Pass@1 results grouped by task difficulty across different approaches. Across DeepSeek-V3, GPT-4o and GPT-5, *ConRAD* consistently achieves higher Pass@1 than all baselines at each difficulty level. On *Easy* issues, *ConRAD* achieves strong Pass@1 that are comparable to or exceed those of prior methods. On *Medium-High* issues, *ConRAD* maintains its effectiveness across three models, achieving Pass@1 of 29.2% with DeepSeek-V3 and 28.5% with both GPT-4o and GPT-5, which outperform all the baselines. The results show that *ConRAD* **helps improve Pass@1 across both easy and more challenging issues**.

*Table 2.* Pass@1 of mini-SWE-agent with and without *ConRAD* on 100 randomly sampled SWE-Bench Lite issues (GPT-5).

| Scaffold | Pass@1(%) |
|---|---|
| mini-SWE-agent | 48.0% (48/100) |
| **mini-SWE-agent + *ConRAD*** | **58.0% (58/100)** ↑ |

**Generalization to a different scaffold.** While our main results use Agentless as the execution scaffold, ConRAD is designed to be scaffold-agnostic (Section 3.4). To verify this empirically, we additionally evaluate ConRAD on mini-SWE-agent, a lightweight variant of SWE-agent with a substantially different agent architecture, on 100 randomly sampled SWE-Bench Lite issues using GPT-5. Adapting ConRAD to mini-SWE-agent(see Appendix H for setup details), we observe that *ConRAD* raises Pass@1 from 48.0% to 58.0% (Table 2), comparable in magnitude to the +10.3% gain on Agentless over the full benchmark. This indicates that *ConRAD*'s gains are not tied to a specific scaffold and transfers across architecturally different agentic APR pipelines.

### 5.2. Ablation Studies

**Effect of the Exemplar Guardian.** We evaluate the effectiveness of the Exemplar Guardian (EG) by comparing the full *ConRAD* system with a variant that disables EG. When EG is disabled, we remove the compatibility filtering step and use the retrieved exemplar directly for plan distillation and inference-time guidance. Removing EG reduces the Pass@1 from 37.7% to 33.7% on GPT-4o, from 40.0% to 37.7% on GPT-5, and from 35.6% to 30.0% on DeepSeek-V3 (Table 3). We further apply the exact McNemar's test (McNemar, 1947) and find that the improvement

*Table 3.* Ablation of Exemplar Guardian (EG): end-to-end Pass@1 of *ConRAD* with and without EG.

| Base Model | Approach | Pass@1 (%) | # Selected Exemplar |
|---|---|---|---|
| GPT-4o | *ConRAD* | 37.7 | 195/300 |
| | *ConRAD* w/o EG | 33.7 (-4.0%) | – |
| GPT-5 | *ConRAD* | 40.0 | 286/300 |
| | *ConRAD* w/o EG | 37.7 (-2.3%) | – |
| DeepSeek-V3 | *ConRAD* | 35.6 | 257/300 |
| | *ConRAD* w/o EG | 30.0 (-5.6%) | – |

*Table 4.* Exemplar Guardian stability across three runs: Agree@3 (exact agreement) and majority vote by group.

| Group | $n$ | Agree@3 | Majority Vote |
|---|---|---|---|
| Transferable | 58 | 53/58 (91.4%) | 58/58 (100%) |
| Not-transferable | 33 | 30/33 (90.9%) | 33/33 (100%) |
| Misleading | 9 | 7/9 (77.8%) | 9/9 (100%) |
| Overall | 100 | 90/100 (90.0%) | 100/100 (100%) |

from including EG is statistically significant ($p$ <0.05). ***This result highlights the importance of exemplar filtering in supporting effective reasoning distillation.***

**Stability of the Exemplar Guardian.** We evaluate the run-to-run stability of Exemplar Guardian (EG) under GPT-4o by executing it three times on 100 stratified samples and measuring label agreements across runs. The sample is stratified by the EG label obtained in our completed experiments (Transferable, Non-transferable, Misleading) to ensure coverage of all three labels. Table 4 reports the results using *Agree@3*, defined as the percentage of decisions whose EG labels are identical across all three runs. Overall, EG achieves 90.0% Agree@3, with per-group agreement of 91.4% on *Transferable*, 90.9% on *Non-transferable*, and 77.8% on *Misleading*. When aggregating three runs by majority vote, all the results remain consistent with the initial run. ***These results indicate that EG provides a stable filtering signal.***

**Reasoning Plan Generation.** We study the effect of the reasoning plan generation using GPT-4o by replacing Backward Reasoning Distillation (BRD) with a forward exploration alternative based on Monte Carlo Tree Search (MCTS) using 100 randomly selected samples. We use a fixed reasoning budget with branching factor 3, and perform 20 rollouts over a 4-level decision process (file localization, function localization, edit localization, patch generation), resulting in *9.71×* more LLM calls and *2.47×* more tokens than BRD per issue on average (more details in Appendix F). Under this controlled MCTS-style configuration, MCTS substantially reduces Pass@1, underperforming both the full *ConRAD* and the Agentless baseline (Table 5). To further isolate the contribution of the reasoning plan from

*Table 5.* Ablation on GPT-4o for (i) Reasoning Plan Generation and (ii) Exemplar Similarity: Pass@1 of *ConRAD* variants.

| Category | Approach | Pass@1 (%) |
|---|---|---|
| Reasoning Plan | *ConRAD* (BRD) | 41.0 |
| | *ConRAD* (MCTS) | 28.0 (-13.0 %) |
| | Exemplar&Patch(no plan) | 32.0 (-9.0 %) |
| | Agentless | 34.0 (-7.0 %) |
| Exemplar Similarity | *ConRAD* (Top-1) | 45.0 |
| | *ConRAD* (Top-3) | 39.0 (-6.0 %) |
| | Agentless | 34.0 (-11.0 %) |

exemplar injection itself, we additionally evaluate an **Exemplar & Patch (no plan)** variant that injects the retrieved exemplar's issue description and ground-truth patch directly into the prompt, without any distilled reasoning plan. This variant underperforms the Agentless baseline despite having access to a similar resolved bug. ***These results suggest that outcome-conditioned backward reasoning distillation can provide more cost-effective guidance than this forward exploration baseline under our fixed budget setting, and the necessity of the distilled reasoning plan.***

**Sensitivity on Exemplar Similarity.** Table 5 shows the impact of exemplar quality on GPT-4o by comparing the top-ranked retrieved exemplar (Top-1) with a lower-ranked candidate (Top-3) across the same 100 random samples. We observe a decrease in the Pass@1 from 45.0% to 39.0%, yet we still outperform the no-retrieval baseline (34.0%). These results indicate that ***stronger exemplar–target alignment increases the benefits of reasoning transfer. More importantly, it also shows that*** *ConRAD* ***remains robust when given moderately imperfect exemplars.***

# 6. Limitations

Our evaluation is conducted on SWE-Bench Lite, a cost-efficient but limited subset of real-world repair tasks that primarily targets Python projects, which may restrict generalization to other languages or domains. In addition, *ConRAD* relies on the availability of suitable historical fixes within the same repository. Projects with sparse bug-fix histories may have fewer effective exemplars. Both the Exemplar Guardian and reasoning distillation depend on LLM-generated judgments, and their effectiveness may vary with the underlying model's reasoning reliability.

# 7. Conclusion

We presented *ConRAD*, a framework for transferring *reusable repair reasoning plans* from historically verified fixes to new program-repair tasks at inference time. *ConRAD* integrates repository-level exemplar mining, conservative exemplar filtering, and reverse-engineered reasoning distillation to reconstruct stage-wise plans that are compat-

ible with standard autoregressive LLMs and can be seamlessly injected into existing APR scaffolds. Crucially, when verified historical fixes exist, reasoning can be *distilled backward* from the known-correct outcome and then reuse to guide future repairs without modifying model parameters. Across three LLM backbones on SWE-Bench Lite, *ConRAD* consistently improves end-to-end repair success and localization accuracy over strong baselines and its execution scaffold. Ablation studies further highlight the importance of exemplar filtering and distilling verified, causally consistent plans. Together, these results suggest that backward distillation from verified outcomes provides an effective and lightweight alternative to expensive forward exploration for scaling LLM-based repository-level program repair.

## Acknowledgements

We acknowledge the support of the Government of Canada's New Frontiers in Research Fund (NFRF), [NFRFE-2024-00612].

## Impact Statement

Although our evaluation shows that *ConRAD* improves the end-to-end Pass@1 and the generated patches can be potentially used to reduce the manual effort taken by developers in fixing bugs, we cannot provide guarantee on the quality of the generated patches. *ConRAD* does not remove the need for standard validation practices, since the final patch is generated by a probabilistic model and must be checked in the target repository (e.g., via test suites and code review) before adoption. Future research is needed to conduct accurate code review of these automatically generated patches. We believe that the open-source nature and the effectiveness of our tool play important roles in making debugging and automated patching more reliable.

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

# A. Appendix:Implementation Details

### A.1. Data Construction.

For each repository $R$ among the 13 SWE-Bench Lite repositories, we construct a repository-specific historical exemplar dataset following the official SWE-Bench data collection protocol. Specifically, we collect all historical issues that have been successfully resolved and are associated with verified pull requests. Each issue report $x$ and its pull request provides a ground-truth patch $y_{patch}$ that passes the project's test suite and therefore serves as a reliable supervision signal for reasoning distillation.

To ensure that each exemplar corresponds to a well-defined and interpretable repair scenario, we apply an additional filtering constraint and retain only issues whose ground-truth patch modifies at most three source files. This constraint serves two purposes: (i) patches touching many files often correspond to refactoring or large-scale changes, making it difficult to attribute the fix to a coherent debugging strategy; and (ii) limiting the number of modified files controls the complexity of the repair task, preserving a clear mapping between the issue and its primary fault locations, which is essential for extracting transferable, step-wise repair reasoning.

From each retained pull request, we extract the modified file paths $y_{file}$ and the functions $y_{function}$ using diff-based analysis. We also record the timestamp $T$ at which the issue was resolved. Formally, each historical exemplar is represented as a structured tuple

$$e = \langle x, y_{file}, y_{function}, y_{patch}, T \rangle.$$

During exemplar retrieval for a target issue, only exemplars with $T$ strictly earlier than the target issue's creation time are considered, preventing temporal leakage.

### A.2. Exemplar Injection into the Repair Pipeline.

*ConRAD* is implemented as an inference-time augmentation on top of an existing repair workflow and does not modify the underlying execution scaffold. In our experiments, we use Agentless as the repair pipeline.

For a given target issue, once a historical exemplar $e$ is accepted by the *Exemplar Guardian*, we inject its distilled reasoning plans into the corresponding stages of the Agentless workflow as structured in-context guidance. Specifically, ConRAD produces three reasoning plans via Reverse-Engineered Reasoning Distillation: a file-level plan $S_{file}$, a function-level plan $S_{func}$, and a patch-level plan $S_{patch}$.

Each reasoning plan the exemplar's issue description $x$ and ground-truth patch $y_k$ are prepended to the prompt of its corresponding stage in the repair pipeline. The file-level reasoning plan $S_{file}$ is injected into the file localization prompt, guiding the model toward relevant source files. Conditional on the predicted file, the function-level plan $S_{func}$ is injected into the function localization prompt to refine the fault location. Finally, the patch-level plan $S_{patch}$ is injected into the patch generation prompt to guide the synthesis of the repair.

All components of *ConRAD* operate at inference time. *ConRAD* does not require fine-tuning of the base language model, nor does it introduce additional environment interactions (e.g., executing tests or invoking external tools). This design ensures that *ConRAD* remains compatible with standard autoregressive LLMs and can be readily applied to other structured repair pipelines beyond Agentless.

# B. Unique Issues Pass@1.

Our intersection analysis is conducted on SWE-Bench Lite under the GPT-4o setting to compare the issues resolved by *ConRAD*. Figure 3 illustrates the overlap of resolved issues between *ConRAD* and four representative baselines: *AutoCodeRover-v2*, *MASAI*, *Moatless Tools*, and *Agentless*. We select these baselines because they are among the strongest GPT-4o baselines and provide per-issue resolution information, enabling a fine-grained overlap analysis. While these methods share a substantial portion of commonly resolved instances, each system also fixes a small set of issues that are not addressed by the others. Notably, *ConRAD* uniquely resolves **10** issues that are missed by all four baselines. This result suggests that *ConRAD* complements existing approaches by resolving a distinct subset of issues beyond those addressed by prior methods.

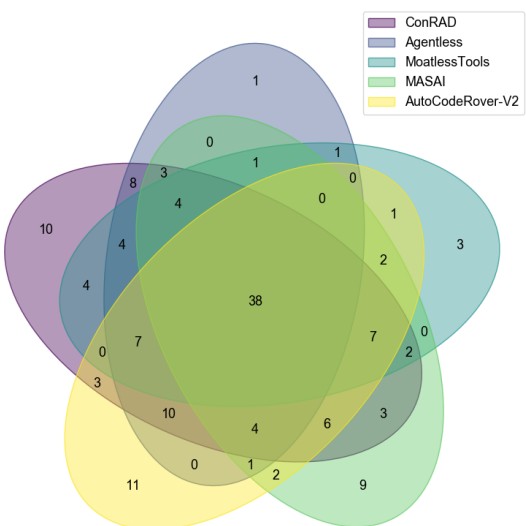

*Figure 3.* Intersection of resolved issues between *ConRAD* and baselines.

## C. LLM-as-a-Judge

Appendix C presents the LLM-as-a-judge prompt used to assess semantic alignment between a target bug issue and candidate historical issues. The prompt implements a structured comparison rubric that evaluates whether a candidate issue exhibits a compatible underlying repair intent with the target issue, and is used to select a single semantically aligned exemplar during the retrieval stage described in Section 3.1.

### LLM-as-a-Judge

```
"""You are an expert debugging assistant with meta-reasoning abilities. Your task is to help a novice
    developer identify the most useful past bug report to assist in locating and fixing the CURRENT
    bug.
You will evaluate 5 candidate bug reports based on **five reasoning strategies**:
1. Structural Similarity: Similar stack traces, error messages, or call graph structure.
2. Module/Component Similarity: Involves the same files, modules, functions, or subsystems.
3. Symptom Similarity: Similar observable behaviors (e.g., button unresponsive, UI freeze).
4. Impact Similarity: Affects the same user flows, APIs, or workflows.
Please follow this process:
- Carefully read the CURRENT bug report.
- For each candidate:
  1. Evaluate it on each of the 5 criteria above (rate 1 to 10).
  2. Select the most helpful report overall.
  3. Provide a short justification for why you chose it as most helpful for fixing and debugging the
      current bug.
Finally, provide your answer in the following strict format using XML-style tags:
- Your selected candidate must be wrapped with <Final choice>CANDIDATE_A/B/C/D/E.</Final choice>
- Your justification must be wrapped with <Justification>...</Justification>
For example:
<Final choice>CANDIDATE_D</Final choice>
<Justification>Candidate D addresses similar components in the 'io.fits' module and has a structural
    similarity to the CURRENT bug, as both involve HDU handling and modifications of the reading
    mechanisms. The error related to variable management in reading HDUs could provide insights into
    ensuring the 'replace' function's expected behavior. The relevance of shared fixes and similar
    testing challenges makes it particularly useful for debugging the CURRENT bug.</JUSTIFICATION>
# CURRENT_BUG_REPORT
{current_bug_report}
---
```

```
# CANDIDATE_BUG_REPORT_A
## Bug Report:
{bug_report_a}
## Fix Description (PR):
{pr_a}
## Fixed Files:
{file_paths_a}
---
# CANDIDATE_BUG_REPORT_B
## Bug Report:
{bug_report_b}
## Fix Description (PR):
{pr_b}
## Fixed Files:
{file_paths_b}
---
# CANDIDATE_BUG_REPORT_C
## Bug Report:
{bug_report_c}
## Fix Description (PR):
{pr_c}
## Fixed Files:
{file_paths_c}
---
# CANDIDATE_BUG_REPORT_D
## Bug Report:
{bug_report_d}
## Fix Description (PR):
{pr_d}
## Fixed Files:
{file_paths_d}
---
# CANDIDATE_BUG_REPORT_E
## Bug Report:
{bug_report_e}
## Fix Description (PR):
{pr_e}
## Fixed Files:
{file_paths_e}
"""
```

## D. Exemplar Guardian

Appendix D details the prompt used by the Exemplar Guardian to assess the semantic compatibility between a retrieved historical issue and a target issue, and to assign a transferability score that determines whether the exemplar is retained for following reasoning distillation.

### Exemplar Guardian

```
"""
You are an expert software debugging analyst specializing in identifying transferable bug-fixing
    patterns.

TASK:
Evaluate this historical issue to determine whether its debugging approach, reasoning pattern, or fix
    strategy can guide solving the CURRENT issue.
We want to include the historical issue as a few-shot example to guide on fixing the current issue.
    Classify each as **Transferable**, **Non-Transferable**, or **Misleading** based on
    transferability and potential to mislead.
```

```
IMPORTANT CALIBRATION:
Historical candidates are retrieved automatically and may be noisy.
Do NOT default to "Transferable".
Default to "Non-Transferable" unless you can point to at least ONE concrete, candidate-specific
    transferable aspect
that clearly connects to the CURRENT issue (root cause / mechanism / component / patch-shape / test
    pattern).

Weak usefulness is allowed, but ONLY when there is concrete evidence from the historical issue.
Generic debugging advice (e.g., "write a regression test", "trace internal path") WITHOUT candidate-
    specific evidence
is NOT sufficient for "Transferable".

<CURRENT_ISSUE>
Issue Summary:
{current_issue_summary.strip()}
</CURRENT_ISSUE>

<HISTORICAL_ISSUES>
{''.join(cand_blocks) if cand_blocks else '(no candidates provided)'}
</HISTORICAL_ISSUES>

-----------------------------------------------------------
EVALUATION CRITERIA (in priority order)
-----------------------------------------------------------

1. **ROOT CAUSE SIMILARITY** *(Highest Priority)*
   - Does the historical issue share the same fundamental problem type?
   - Examples: null pointer, race condition, off-by-one error, resource leak, logic inversion, API
       misuse, state management issue.

2. **CAUSAL CHAIN TRANSFERABILITY**
   - Does the historical issue demonstrate a similar sequence of events leading to failure?
   - Can the diagnostic reasoning path be reused effectively?
   - If both the historical and current issues involve a system's automated or implicit behavior
       suppressing or overriding explicit developer intent, treat the pair as potentially transferable,
        even if they occur in different subsystems or technologies.
       This includes cases where:
       Automatically generated logic, caching, schema evolution, or reflection-based mechanisms ignore,
           replace, or bypass explicit user configuration or overrides.
       A newer system version introduces hidden precedence or dispatch changes that silently alter
           user-visible behavior.
       The debugging process required inspecting generated code paths, internal dispatch layers, or
           meta-level logic to restore expected control flow.

3. **FIX STRATEGY APPLICABILITY**
   - Is the general solution approach (not necessarily the code) relevant?
   - Examples: adding validation, reordering operations, caching, defensive copying, changing data
       structures.

4. **CONTEXTUAL ALIGNMENT**
   - Are the system components, architectural layers, or runtime contexts aligned?
   - Examples: UI layer, database access, API handling, concurrency, initialization logic.

5. **DEBUGGING TECHNIQUE VALUE**
   - Does the historical issue reveal useful diagnostic or investigation methods?
   - Examples: specific test cases, reproduction steps, or diagnostic heuristics.

-----------------------------------------------------------
DECISION GUIDELINES
-----------------------------------------------------------
```

```
### Mark **Transferable** ONLY if:
- You can extract >= 1 concrete, candidate-specific transferable aspect grounded in the historical
     issue AND
  it matches the current issue in at least ONE of the following:
  (1) root cause similarity, OR
  (2) explicit shared mechanism pattern (e.g., descriptor/property/metaclass dispatch, caching/registry
        precedence) mentioned in BOTH,
  (3) component/layer overlap (same module family / same runtime layer / same API boundary),
  (4) a clearly reusable patch-shape that applies to the same type of bug (e.g., precedence ordering,
        guard condition) with evidence.

Workflow-only transfer (repro -> locate -> minimal fix) WITHOUT any of (1)-(4) is NOT sufficient for
     Transferable.
In that case, label as "Non-Transferable" with usefulness_score in (0.0, 0.1].

### Mark **Misleading** if:
- The candidate would likely push toward a SPECIFIC wrong fix direction/layer/module (e.g., schema
     editor changes for a runtime dispatch bug),
  OR suggests a misleading patch category that wastes time.
(Mere subsystem difference is NOT Misleading.)

### Mark **Not useful** if:
- You cannot name ANY actionable transferable aspect (where-to-look / what-to-trace / what-to-test /
     patch-shape).
- No shared mechanism pattern can be articulated.
- The issue is too vague or unrelated such that it offers neither workflow nor diagnostic reuse.

------------------------------------------------------------
CLASSIFICATION PRIORITY & SCORING
------------------------------------------------------------

- If both *Transferable* and *Misleading* aspects exist -> **Mark "Misleading"** (safety first).
- If uncertain between *Misleading* and *Non-Transferable*, consider whether the mismatch could
     mislead.
  - If yes -> **Misleading**.
  - If truly neutral -> **Non-Transferable**.

**Transferability Score Range:**
- Misleading: [-1.0, -0.1]
  - <= -0.5 when the historical bug belongs to a completely different subsystem and could mislead
       debugging.
- Not Transferable: (-0.1, 0.1)
- Transferable: [0.1, 1.0]

------------------------------------------------------------
OUTPUT FORMAT *(strict JSON only)*
------------------------------------------------------------

{{
  "candidates": [
    {{
      "idx": <int>,
      "id": "<original_id>",
      "decision": "Transferable" | "Non-Transferable" | "Misleading",
      "usefulness_score": <float between -1.0 and 1.0>,
      "confidence_score": <float between 0.0 and 1.0>,
      "transferable_aspects": [<list of 1 to 3 concrete items: for Transferable -> what transfers; for
           Misleading -> what misleads; for Non-Transferable -> can be empty array>],
      "reason": "<1 to 2 concise sentences explaining WHY, referencing the above criteria. For
           Misleading, explicitly state what could mislead.>"
    }}
  ]
```

```
    }}

    -----------------------------------------------------------
    CRITICAL RULES
    -----------------------------------------------------------
    - EVIDENCE RULE: Every item in "transferable_aspects" must be grounded in the historical issue content
      (e.g., mentions a specific mechanism, module/layer, failure mode, test pattern, or patch shape).
      If you cannot ground it, do NOT include it and do NOT label the candidate as Transferable based on it
          .
    - Before choosing "Non-Transferable", attempt to find a candidate-specific transferable aspect.
      If you cannot ground any aspect in the historical issue, explicitly state "no grounded transferable
          aspect".
    - Evaluate **every** candidate [1..N]; return empty array if none.
    - Output **valid JSON only** no Markdown or extra text.
    - Be **specific** in reasoning; vague or abstract explanations reduce utility.
    - Emphasize **root cause + context alignment** over superficial symptom overlap.
    - Be liberal in identifying potentially **Transferable** reasoning but conservative about **Misleading
        ** ones.
    """.
```

### Example: Exemplar Guardian Self-reflection

```
You are performing critical self-reflection on your previous judgment. Be honest about potential
    errors and focus on improving accuracy.

<CURRENT ISSUE>
Issue Summary: {issue_summary}
</CURRENT_ISSUE>

<HISTORICAL_ISSUES>
Issue Summary: {issue_summary}
</HISTORICAL_ISSUES>

<PREVIOUS JUDGMENT>
### The decision: {decision}
### The Transferability score: {score}
### The confidence score: {confidence_score}
### The transferable aspects: {aspects}
### The reason: {reason}
</PREVIOUS JUDGMENT>
```

## E. Backward Reasoning Distillation

Appendix E presents the prompts used for Stage 3 (Backward Reasoning Distillation), which consists of two steps: initial plan construction and stage-wise refinement. The corresponding prompts are provided in Appendix E.1 and Appendix E.2, respectively.

### E.1. Prompt for Initial Plan Construction

### Sub-task: File Localization Prompt

```
You are an expert software engineer tasked with locating the most relevant file for fixing a bug.

Please look through the following GitHub problem description and Repository structure and provide a
    list of files that one would need to edit to fix the problem.

You are given:
```

- A GitHub Issue Description.
- A snapshot of the project Project Directory Structure (program file paths only).
- The actual file that was modified to fix this GitHub Issue (ground truth).

Your task is to **reason step-by-step** how to narrow down the possible files **starting from the
    entire directory structure**, and explain why the ground truth file is the most appropriate for
    this bug fix.

Avoid vague jumps. Instead, explain your reasoning like a detective narrowing down suspects.

---

<GitHub Issue Description>
{problem}
</GitHub Issue Description>

<Project Directory Structure>
{repository_structure}
</Project Directory Structure>

<Ground Truth Modified Files>
{gt_files}
</Ground Truth Modified Files>

---
Step-by-step reasoning:
1. First, I look at the bug report to extract key terms, affected modules, and clues.
2. Then, I scan the directory structure to find files or directories whose names or paths semantically
    relate to those clues.
3. Among those, I consider the functionality suggested by the bug (e.g., GPU, utils, tensors).
4. Based on likely responsibilities and location of logic, I narrow it down further.
5. I find that the files "{gt_files}" are the best match because...

Please provide the full step by step reasoning, and the full path and return at most 5 files.
The returned files should be separated by new lines ordered by most to least important.
For example:
```
file1.py
file2.py
```
Return the location(s) wrapped with ```
Your reasoning should start with "### Thinking:", and your answer should start with "### Answer:".

## Sub-task: Function Localization Prompt

You are an expert software engineer tasked with locating the most relevant locations for fixing a bug.

Please look through the following GitHub Problem Description and the Skeleton of Relevant Files.
Identify all locations that need inspection or editing to fix the problem, including directly related
    areas as well as any potentially related global variables, functions, and classes.
For each location you provide, either give the name of the class, the name of a method in a class, the
    name of a function, or the name of a global variable.

You are given:
- A GitHub Problem Description.
- A snapshot of the Skeleton of Relevant Files.
- The actual locations were modified to fix this bug (ground truth), which can be the name of the
    class, the name of a method in a class, the name of a function, or the name of a global variable.

Your task is to **reason step-by-step** how to narrow down the possible locations**starting from the
    entire Skeleton of Relevant Files**, and explain why the ground truth locations are the most

```
        appropriate for this bug fix.

Avoid vague jumps. Instead, explain your reasoning like a detective narrowing down suspects.

---

## GitHub Problem Description:
{problem}

## Skeleton of Relevant Files:
{file_skeleton}

## Ground Truth Locations:
# The following entities have been identified as **very likely candidates** for being involved in the
    issue. Beyond your standard reasoning, please give special attention to the following entities, as
     they are very likely to be relevant to the issue.
{gt_related_elements}

---

Step-by-step reasoning:
1. First, I look at the bug report to extract key terms, affected modules, and clues.
2. Then, I scan the Skeleton of Relevant Files to find classes, functions, or directories whose names
    or paths semantically relate to those clues.
3. Among those, I consider the functionality suggested by the bug.
4. Based on likely responsibilities and locations of logic, I narrow it down further.
5. I find that the locations "{gt_related_elements}" are the best match because...

Now explain your full reasoning step-by-step, but do not start with conversational phrases such as '
    Sure' in the response.And please provide the complete set of locations as either a class name, a
    function name, or a variable name.
Note that if you include a class, you do not need to list its specific methods. You can include either
     the entire class or don not include the class name and instead include specificmethods in the
    class.
For example:
```
full_path1/file1.py
function: my_function_1
class: MyClass1
function: MyClass2.my_method

full_path2/file2.py
variable: my_var
function: MyClass3.my_method

full_path3/file3.py
function: my_function_2
function: my_function_3
function: MyClass4.my_method_1
class: MyClass5
```
Return the location(s) wrapped with ```
Your reasoning should start with "### Thinking:", and your answer should start with "### Answer:".
```

## Sub-task: Patch Generation

```
We are currently solving the following issue within our repository. Here is the issue text:
--- BEGIN ISSUE ---
{problem}
--- END ISSUE ---
```

Below are some code segments, each from a relevant file. One or more of these files may contain bugs.
--- BEGIN FILE ---
```
{file_contents}
```
--- END FILE ---

### Additional Contextual Clues:
The following *SEARCH/REPLACE* edit have been identified as **the ground truth** for being involved in
    the issue. Beyond your standard reasoning, please give special attention to the following *SEARCH
    /REPLACE* edit.
{search_replace}

---

Step-by-step reasoning:
1. I begin by analyzing the bug report to identify the expected behavior and the faulty or missing
    logic.
2. Then, I locate the relevant file and function or class by reading through the provided code
    segments.
3. I identify the exact lines that cause the bug or where the fix needs to be inserted (e.g., a
    missing condition, incorrect return, or broken logic).
4. I select a contiguous code block from the current code that I would like to change this is the *
    SEARCH* block.
5. I then write the correct version of the code as the *REPLACE* block, ensuring all indentation and
    spacing is preserved exactly.
6. I format the result in the required SEARCH/REPLACE format and wrap it in a code block.

Please first localize the bug based on the issue statement, and then generate *SEARCH/REPLACE* edits
    to fix the issue.

Every *SEARCH/REPLACE* edit must use this format:
1. The file path
2. The start of search block: <<<<<<< SEARCH
3. A contiguous chunk of lines to search for in the existing source code
4. The dividing line: =======
5. The lines to replace into the source code
6. The end of the replace block: >>>>>>> REPLACE

Here is an example:

```python
### mathweb/flask/app.py
<<<<<<< SEARCH
from flask import Flask
=======
import math
from flask import Flask
>>>>>>> REPLACE
```

Please note that the *SEARCH/REPLACE* edit REQUIRES PROPER INDENTATION. If you would like to add the
    line 'print(x)', you must fully write that out, with all those spaces before the code!
Wrap the *SEARCH/REPLACE* edit in blocks ```python...```

## E.2. Prompt Template for Stage-wise Refinement

**Candidates Generation for File Localization Reasoning Plan**

```
"""You are an expert at file localization for bug fixing tasks.

## Task Description:
You need to locate the most relevant files that need to be edited to fix a bug, based on the issue
    description and repository structure.

## GitHub Issue:
{problem}

## Repository Structure:
{stage_context.get('repository_structure', 'Not provided')}

## Context Before This Step:
{context_before if context_before else "This is the first step."}

## Current Step {step_number}:
{step_content}

## Context After This Step:
{context_after if context_after else "This is the last step."}

---

Please generate {num_variants} improved variants of Step {step_number} ONLY.
Each variant should:
1. Provide more specific reasoning about file locations based on the repository structure
2. Explain WHY certain files/directories are relevant to the issue
3. Use technical terminology related to the project structure
4. Maintain consistency with surrounding steps

Each variant should start with "Step {step_number}:" and focus on file localization reasoning.

Format your response as:
### Variant 1:
[your improved step here]

### Variant 2:
[your improved step here]

### Variant 3:
[your improved step here]
"""
```

**Candidates Generation for Function Localization Reasoning Plan**

```
"""You are an expert at function/class localization for bug fixing tasks.

## Task Description:
You need to locate the specific functions, classes, or methods that need to be inspected or modified
    to fix a bug, based on the file skeleton.

## GitHub Issue:
{problem}

## File Skeleton (Classes and Functions):
{stage_context.get('file_skeleton', 'Not provided')}
```

```
## Context Before This Step:
{context_before if context_before else "This is the first step."}

## Current Step {step_number}:
{step_content}

## Context After This Step:
{context_after if context_after else "This is the last step."}

---

Please generate {num_variants} improved variants of Step {step_number} ONLY.
Each variant should:
1. Provide specific analysis of which functions/classes/methods are relevant
2. Explain the relationships between components
3. Reference the file skeleton structure
4. Maintain consistency with surrounding steps

Each variant should start with "Step {step_number}:" and focus on function/class localization
    reasoning.

Format your response as:
### Variant 1:
[your improved step here]

### Variant 2:
[your improved step here]

### Variant 3:
[your improved step here]
"""
```

## Candidates Generation for Patch Generation Reasoning Plan

```
"""You are an expert at generating code patches for bug fixing tasks.

## Task Description:
You need to generate the exact SEARCH/REPLACE patch to fix a bug, based on the file contents.

## GitHub Issue:
{problem}

## File Contents:
{stage_context.get('file_content', 'Not provided')}

## Context Before This Step:
{context_before if context_before else "This is the first step."}

## Current Step {step_number}:
{step_content}

## Context After This Step:
{context_after if context_after else "This is the last step."}

---

Please generate {num_variants} improved variants of Step {step_number} ONLY.
Each variant should:
1. Provide clearer reasoning about the code changes needed
2. Explain the logic behind the fix
3. Reference specific code patterns or structures
```

4. Maintain consistency with surrounding steps

Each variant should start with "Step {step_number}:" and focus on patch generation reasoning.

Format your response as:
### Variant 1:
[your improved step here]

### Variant 2:
[your improved step here]

### Variant 3:
[your improved step here]
"""

## Step Comparison and Selection Prompt

"""You are an expert evaluator of reasoning quality for software engineering tasks.

## Problem Description:
{problem}

## Ground Truth outcome:
{ground_truth}

## Original Step {step_number}:
{original_step}

## Variant 1:
{variants[0] if len(variants) > 0 else "N/A"}

## Variant 2:
{variants[1] if len(variants) > 1 else "N/A"}

## Variant 3:
{variants[2] if len(variants) > 2 else "N/A"}

---

Please evaluate each version (Original, Variant 1, Variant 2, Variant 3) based on:
1. Consistency with the surrounding reasoning plan context.
2. Compatibility with the verified outcomes Y, especially the final patch y.
3. Specificity and actionability that support downstream localization or patch generation.

For each version, provide a score from 1-10 and brief justification.
Then select the BEST version overall.

Format your response as:
### Original:
Score: X/10
Justification: [brief explanation]

### Variant 1:
Score: X/10
Justification: [brief explanation]

### Variant 2:
Score: X/10
Justification: [brief explanation]

### Variant 3:

```
  Score: X/10
  Justification: [brief explanation]

  ### Best Version: [Original/Variant 1/Variant 2/Variant 3]
  Reason: [why this is the best]
  """
```

## F. MCTS Budget and Token Overhead

We report the reasoning budget and inference-time overhead of the MCTS-based forward-search baseline used in Table 6. We measure overhead by the number of LLM calls and the total generated tokens per issue (input+output), averaged over the same evaluation set. This overhead comparison is measured under our MCTS-based forward-search baseline (N=100) and is intended to characterize the cost of search-style refinement in this controlled setting, rather than a system-level comparison to all MCTS-based APR systems.

**Budget configuration.** We apply MCTS over a 4-level repair decision process (file-level localization, function-level localization, edit-location localization, patch generation) with branching factor $B=3$ and $R=20$ rollouts. We cap the search depth to the 4 decision levels and use the same base model and prompting scaffold as other variants. We emphasize that this configuration serves as a controlled forward-search baseline for ablation; different MCTS-style systems may adopt different action spaces, priors, and budget schedules, which can change both overhead and performance.

**Token and call overhead.** Table 6 summarizes the average LLM calls and token usage per issue. We report both input and output tokens, as well as total tokens (input+output). We also report the relative overhead compared to BRD. Note that Table 6 reports raw overhead statistics; a matched-budget effectiveness comparison would require explicitly constraining search to the same call/token budget and is left for future work.

*Table 6.* Inference-time overhead of MCTS vs. BRD. "Calls/issue" counts the number of LLM invocations; tokens are averaged per issue and include both input and output tokens.

| Method | Average calls | ×BRD (calls) | Average token. | ×BRD (tokens) |
|---|---|---|---|---|
| BRD (ours) | 24.7 | 1.00 | 14.3 k | 1.00 |
| MCTS | 240 | 9.71 | 35.4 k | 2.47 |

## G. Compatibility with Other Agentic APR Pipelines

*ConRAD* can be applied to generic APR agents. Here, we describe how *ConRAD* generates reasoning plans $S^*$ when SWE-agent or mini-SWE-agent are used as the execution scaffold (Yang et al., 2024). Following SWE-agent's standard trajectory format, the reasoning plan $S^*$ provided by *ConRAD* consists of a complete *trajectory demonstration* composed of alternating Thought/Action/Observation turns (See in Figure 4).

**SWE-Agent Trajectory Format**

```
{
  "Response": "# This is the output of the LM",
  "Thought": "# Then parse it into thoughts",
  "Action": "Parse it into actions",
  "Observation": "# And execute the action, resulting in the output",
  ...
}
```

*Figure 4.* SWE-agent official trajectory format (SWE-agent Team, 2025).

*ConRAD* **for SWE-agent/mini-SWE-agent.** Since the Stages 1 and Stage 2 of *ConRAD* are agent-agnostic (retrieval and filtering operate purely over historical issues), we primarily explain how Stage 3 generate refined reasoning plan and inject it into downstream agents. To produce guidance that is directly consumable by SWE-agent, *ConRAD* distills an agent-native `Thought/Action/Observation` demonstration from a historical exemplar with a verified fix. Given the exemplar issue and repository context, *ConRAD* first elicits an *initial* reasoning plan $S'$ that an agent could plausibly follow to solve the exemplar, capturing the high-level decision path from localization to patch drafting in the same interaction format used by SWE-agent. Crucially, *ConRAD* does not merely provide a post-hoc explanation of the verified patch. Instead, it performs *reverse-engineered reasoning distillation with step-wise refinement*: conditioning on the exemplar's verified supervision signals (ground-truth bug location and the final patch), *ConRAD* iteratively rewrites each turn's `Thought` so that the resulting trajectory is causally consistent with the verified outcome. This backward refinement removes spurious exploration and irrelevant branches, and sharpens actionable decision logic that an agent can reuse. The final product is a SWE-agent-compatible reusable reasoning plan $S^*$ that distilled from historically verified fixes into `Thought/Action/Observation` turns, enabling prompt-level injection to guide future target issues.

**Inference time usage.** At inference time, we prepend the distilled exemplar demonstration $S^*$ as the few-shot *Demonstration* prefix in SWE-agent's prompt (before the target *Issue statement*). Optionally, a brief reminder summarizing key decisions from $S^*$ can be re-attached at each iteration (or at different stages such as localization and editing) to mitigate context drift, without modifying the agent's core algorithm.

## H. Empirical Validation on mini-SWE-agent

To verify that *ConRAD*'s gains generalize beyond Agentless, we evaluate *ConRAD* on mini-SWE-agent, a lightweight variant of SWE-agent with a substantially different agent architecture.

**Setup.** We sample 100 issues uniformly at random from SWE-Bench Lite (random seed fixed for reproducibility) and run both mini-SWE-agent and mini-SWE-agent + *ConRAD* on this same subset using GPT-5 as the base model. For each target issue, the first two stages of *ConRAD* (Repository-Level Exemplar Mining and Exemplar Guardian) operate identically to the Agentless setting. Stage 3 (Backward Reasoning Distillation) is adapted to produce a Thought/Action/Observation demonstration trajectory in mini-SWE-agent's native format from the retrieved exemplar's verified fix, following the procedure described above (Section G). The distilled trajectory is prepended to mini-SWE-agent's prompt as a few-shot demonstration prefix; the agent's core control loop, action space, and step budget are unchanged. When the Exemplar Guardian rejects the retrieved exemplar, the mini-SWE-agent baseline is invoked without injection.

**Results.** Table 2 (in Section 5.1) reports Pass@1 with and without *ConRAD*. Adopting *ConRAD* on mini-SWE-agent improves Pass@1 from 48.0% (48/100) to 58.0% (58/100), a +10.0 absolute improvement that is comparable in magnitude to the +10.3% gain observed on Agentless over the full benchmark. This supports the claim that *ConRAD*'s benefits arise from outcome-conditioned reasoning distillation itself, rather than from any property specific to the Agentless scaffold.

