# OpenReview forum: "From Historical Patches to Repair Plans: Outcome-Conditioned Reasoning for Repository-Level Program Repair"
_ICML.cc/2026/Conference — ICML 2026 regular_

### Official Review · Reviewer_AHzo · 2026-03-06

**Soundness:** 2
**Presentation:** 3
**Significance:** 3
**Originality:** 3
**Overall Recommendation:** 4
**Confidence:** 4

**Summary:**

This paper introduces ConRAD, a framework designed to enhance APR agents by distilling reasoning trajectories from historical fixes. The approach follows a three-stage pipeline: (1) retrieving relevant patches from the same repository, (2) leveraging LLMs to generate reasoning trajectories constrained by known ground truths, and (3) using a trajectory as one-shot prompts to guide APR agents in resolving new issues. The authors evaluate ConRAD across SWE-Bench Lite, demonstrating that the framework significantly boosts the performance of baseline APR agents.

**Compliance With Llm Reviewing Policy:**

Affirmed.

**Final Justification:**

I appreciate the authors' effort in providing a thorough reply. Since my original concerns have been fully resolved, I have raised my score from 3 to 4.

**Key Questions For Authors:**

See the weakness above.

**Limitations:**

yes

**Strengths And Weaknesses:**

Strength:

1. The authors provide an extensive empirical evaluation. The inclusion of Pass@1, %File, and %Func across various baseline APR agents effectively demonstrates the framework's performance gains.

2. The paper is well-structured, particularly in its description of the ConRAD workflow. The transition from historical patch retrieval to reasoning distillation is logically presented, making the framework's implementation details accessible and reproducible.

Weakness:

1. The authors argue that ConRAD addresses the "outcome uncertainty" of existing forward-search approaches by imposing a global outcome constraint (i.e., including the ground-truth patch in the prompt). However, existing forward-search methods often employ their own filtering or verification mechanisms to evaluate the correctness of generated trajectories. Please clarify the fundamental difference between ConRAD’s grounding and existing trajectory filtering methods. Furthermore, would an ablation study comparing ConRAD’s ground-truth injection against a standard "generate-and-filter" baseline be possible to demonstrate the specific value of this constraint?

2. While the paper claims compatibility with various APR scaffolds, the evaluation is heavily centered on the Agentless+ConRAD configuration. This narrow focus makes it difficult to determine whether the observed improvements are a result of a synergistic effect unique to the Agentless architecture or whether ConRAD provides a consistent lift across more diverse agentic frameworks.

3. Table 1 shows a notable discrepancy: while GPT-4o shows significant gains across all metrics, the improvements for GPT-5 and DeepSeek-V3 are minimal in %File and %Func metrics, even though Pass@1 remains high. The authors should provide a root-cause analysis or a qualitative case study explaining why more advanced models do not benefit as much from ConRAD.

4. The results indicate that Agentless+ConRAD (MCTS) actually underperforms compared to the Agentless baseline. This raises questions about the necessity of the reasoning distillation stage. Could the authors evaluate a baseline where the historical issue/patch information is injected directly into the APR agent’s prompt without the sequence of reasoning plan? This would help isolate whether the reasoning plan itself is the catalyst for improvement.

5. In the Exemplar Guardian stage, if a selected historical issue is discarded, how does the framework proceed? Does ConRAD automatically fall back to the next highest-ranked candidate from the top-5 retrieved in the Repository-Level Exemplar Mining stage, or is that specific repair attempt abandoned?

---

> ### Author Rebuttal · Authors · 2026-03-31
>
> Thank you for the feedback and for taking the time to review our paper. We conducted additional experiments using a **new agent scaffold** (mini-swe-agent), a new ablation study using the **raw issue/patch**, and clarified other confusions. Please find the responses below.
>
>
>
> >### W1: Fundamental difference between ConRAD's ground-truth constraint and existing trajectory filtering methods
>
> **Fundamental difference.**  The key distinction is when outcome information is introduced. Existing forward-search methods apply verification after candidate trajectories have been generated, whereas ConRAD conditions on the verified patch during reasoning construction. In forward-search approaches, trajectories are sampled without knowing whether they will succeed and are then filtered by test execution or model scoring. Such filtering can improve selection, but it does not constrain how the intermediate reasoning is produced: the steps may still be locally plausible yet globally inconsistent with the final correct fix. In contrast, BRD uses the verified patch as part of the generation context, so each reasoning step is constructed to remain compatible with a known-correct outcome from the outset. Thus, ConRAD is not simply adding another filtering mechanism; it replaces post hoc trajectory selection with outcome-conditioned plan distillation.
>
> **On the proposed ablation.**  Our MCTS ablation (Table 4\) already provides a controlled comparison in this spirit. Using the same exemplar, the MCTS variant requires 9.71× more LLM calls yet achieves only 28.0% Pass@1, below even the no-retrieval Agentless baseline (34.0%), whereas BRD reaches 41.0%. These results indicate that forward exploration followed by selection is not equivalent to conditioning reasoning on a verified outcome, and that BRD provides more effective and more computationally efficient guidance in our setting.
>
>
>
> >### W2: Evaluation centered on Agentless; unclear if improvements transfer to other scaffolds.
>
> We conducted an additional experiment using mini-swe-agent, a lightweight variant of SWE-agent, on 100 randomly sampled issues with GPT-5. We adapted ConRAD to the mini-swe-agent’s formant. The table below shows the result. **Adopting ConRAD to mini-swe-agent improves Pass@1 by over 10% (48% to 58%)**, similar to Agentless (10.3%). The finding shows that ConRAD also provides benefits to other scaffolds with a different architecture.
> | Scaffold | Pass@1 |
> |---|---:|
> | mini-swe-agent (baseline) | 48.0% (48/100) |
> | mini-swe-agent + ConRAD | 58.0% (58/100) |
>
>
>
> >### W3: GPT-5 and DeepSeek-V3 show minimal %File/%Func gains despite high Pass@1.
>
> This is because the baseline already achieves high localization accuracy.  GPT-5 reaches 85.0% in %File and 69.0% in %Func, leaving limited room for improvement. The more important observation is that **the bottleneck has shifted from localization to patch generation**: although the GPT-5 baseline correctly localizes the target function in 69.0% of cases, it resolves only 29.7% of issues. Thus, for stronger models, limited improvement in %File/%Func does not indicate limited value, but shows that fault localization is no longer the main constraint.
>
>
> >### W4: Is the reasoning plan necessary? Ablation of direct issue/patch injection without reasoning distillation
>
> We have conducted this ablation: we inject the historical exemplar's issue description and ground-truth patch directly into the APR agent's prompt as a one-shot demonstration, without the BRD-generated reasoning plan.
>
> | Variant | Pass@1 |
> |---|---:|
> | Agentless (baseline) | 34.0% |
> | + exemplar issue/patch only (no plan) | 32.0% |
> | + MCTS (forward reasoning) | 28.0% |
> | + BRD (backward reasoning) | 41.0% |
>
>
> This result shows that directly injecting the historical issue/patch is insufficient for effective transfer, and **the result is worse than the Agentless baseline**. ConRAD shows that the model benefits not from just "seeing a similar problem and its answer," but from the reasoning that explains how the diagnosis and repair proceed.
>
>
> >### W5: When the Guardian rejects a candidate, does ConRAD fall back to the next candidate or abandon exemplar guidance entirely?
>
> ConRAD falls back to unmodified Agentless rather than cascading to the next candidate. Since the top-1 is already the most semantically aligned exemplar selected by the LLM-based judge, lower-ranked candidates are less likely to pass the Guardian's transferability criteria. In practice, this affects few cases: the Guardian accepts 286/300 exemplars with GPT-5 and 257/300 with DeepSeek-V3 (Table 2). Cascading is a natural extension that could improve coverage, and we plan to explore it in future work.

---

> > ### Author Rebuttal · Reviewer_AHzo · 2026-04-01
> >
> > I appreciate the authors' effort in providing a thorough reply. Since my original concerns have been fully resolved, I have raised my score from 3 to 4.

---

> > > ### Author Response · Authors · 2026-04-03
> > >
> > > We sincerely thank the reviewer for the careful evaluation and for raising the score. We are glad that our rebuttal was able to fully address the original concerns. We will incorporate the suggestions discussed during the review process into the final version to further improve the paper. Thank you again for the valuable feedback that helped strengthen our work.

---

### Official Review · Reviewer_6euA · 2026-03-09

**Soundness:** 3
**Presentation:** 3
**Significance:** 3
**Originality:** 3
**Overall Recommendation:** 4
**Confidence:** 4

**Summary:**

The authors propose Conditional Reasoning Distillation (ConRAD), a framework designed to address the fragility and high overhead of long-horizon repository-level automated program repair (APR) by leveraging successful reasoning patterns from historically resolved issues. To avoid the uncertainty and compute-intensiveness of traditional forward exploration, ConRAD identifies semantically aligned in-repository exemplars, filters them through a conservative Exemplar Guardian to prevent negative transfer, and distills structured repair plans by reconstructing reasoning backward from verified ground-truth patches. Evaluated on SWE-Bench Lite, ConRAD improves Pass@1 by up to +10.4% across various LLM backbones.

**Compliance With Llm Reviewing Policy:**

Affirmed.

**Final Justification:**

While the response to does not fully quantify the impact of limited historical exemplars and evaluation on other languages is still unclear enough, the clarifications are sufficient overall. I will keep my recommendation as Weak Accept.

**Key Questions For Authors:**

1. How does the performance of ConRAD degrade as the number of available historical exemplars in a repository decreases?

2. Do the authors have plans to evaluate the framework on statically typed languages (like C++ or Java) where the "reasoning plans" might need to account for stricter architectural constraints?

**Limitations:**

yes

**Strengths And Weaknesses:**

### **Strengths**


1. Good Concept:

The idea of Backward Reasoning Distillation (BRD) is a good use of historical data. Instead of asking the model to find a needle in a haystack, it knows the correct patch beforehand to reconstruct the efficient logical path to that answer.


2. Good Empirical Results:

The framework demonstrates good performance gains across multiple state-of-the-art LLMs. On the SWE-Bench Lite benchmark, it improved Pass@1 scores by 10.4% for GPT-4o, 8.6% for DeepSeek-V3, and 10.3% for GPT-5.



### **Weaknesses**


1. Dataset and Repository Dependency:

The evaluation is limited to a single dataset (SWE-Bench Lite), which focuses primarily on Python projects. Furthermore, the system is heavily dependent on the existence of a robust historical fix database; repositories with sparse histories or "cold start" projects may see limited benefits.



2. High Pre-processing Overhead:

Although the authors emphasize efficiency over search-based methods like MCTS, the distillation process itself is quite heavy. Each "plan" requires an average of 24.7 LLM calls per issue to generate and refine, which represents a significant offline compute cost compared to zero-shot approaches.

---

> ### Author Rebuttal · Authors · 2026-03-31
>
> Thank you very much for the feedback. Please find our responses below.
>
>
> >### Q1. How does the performance of ConRAD degrade as the number of available historical exemplars in a repository decreases?
>
> When no suitable exemplar is found, the Exemplar Guardian falls back to the unmodified repair agent (e.g., Agentless or mini-swe-agent). In Table 2, we evaluated the effectiveness of the Exemplar Guardian comparing ConRAD with and without the Guardian. Across models, adding the Guardian improves Pass@1 by 2.3%–5.6% over removing it.
>
>
> We conducted a preliminary analysis on repositories with different scales of eligible historical issues (see table below). Django (1,032 eligible historical issues) sees \+9.4%/+5.5% improvement over the baseline, while Sympy (only 230\) still shows \+7.5%/+12.5% (GPT-4o/GPT-5). While most issues may not have a matching historical issue, ConRAD falls back to the original agent scaffold but improves those with a matching issue. Nevertheless, we believe this is an interesting area and plan to explore cross-repository plan transfer as future work.
>
> | Repo | Hist. Issues | In Lite | GPT-4o (New resolved / Impr.) | GPT-5 (New resolved / Impr.) |
> | :---: | :---: | :---: | :---: | :---: |
> | Django | 1,032 | 127 | \+12 / \+9.4% | \+7 / \+5.5% |
> | Matplotlib | 663 | 18 | \+3 / \+16.7% | \+4 / \+22.2% |
> | Sympy | 230 | 80 | \+6 / \+7.5% | \+10 / \+12.5% |
>
>
> >### Q2 & W1. Plans to evaluate on statically typed languages (C++, Java) where reasoning plans may need to account for stricter constraints.
>
> Yes. ConRAD is not Python-specific: BRD reasons over issue descriptions, code diffs, and repair outcomes at the level of repair intent rather than language-specific syntax. This makes statically typed languages such as C++ and Java natural extension settings, and SWE-bench Multilingual is a natural next-step benchmark.
>
>
>
> >### W2: Significant Distillation Overhead Compared to Zero-Shot Approaches
>
> We agree that ConRAD introduces additional offline cost compared to zero-shot prompting. However, unlike search-based methods that spend compute on every new issue at inference time, ConRAD uses this offline compute once to **distill a reasoning plan that can later be saved and reused**. In our experiments, this cost gives great returns: under GPT-4o, ConRAD improves Pass@1 by 10.4 absolute percentage in Pass@1 over zero-shot Agentless. It is also much cheaper than MCTS, which requires 9.71 times more LLM calls than ConRAD, while achieving 13 points lower in Pass@1. Moreover, a one-shot variant that injects only the exemplar issue and patch, without reasoning distillation, reaches 32%, slightly below the 34% zero-shot baseline, showing that the gain comes from the distilled plan itself rather than exemplar injection alone. Importantly, **the distilled plan can be saved and reused across future matched issues, so the offline cost can be amortized over time**.

---

> > ### Author Rebuttal · Reviewer_6euA · 2026-04-01
> >
> > Thanks for the rebuttal. While the response to does not fully quantify the impact of limited historical exemplars and evaluation on other languages is still unclear enough, the clarifications are sufficient overall. I will keep my recommendation as Weak Accept.

---

> > > ### Author Response · Authors · 2026-04-03
> > >
> > > We sincerely appreciate your thorough and constructive review. The insightful suggestions have been very helpful in improving the quality of our paper, and we will continue to refine the manuscript accordingly.

---

### Official Review · Reviewer_EuHZ · 2026-03-09

**Soundness:** 3
**Presentation:** 3
**Significance:** 3
**Originality:** 3
**Overall Recommendation:** 4
**Confidence:** 4

**Summary:**

This paper introduces ConRAD (Conditional Reasoning Distillation), an inference-time framework designed to improve repository-level automated program repair (APR). The authors address a core challenge: while previous APR systems often treat each software bug as an isolated problem, real-world repositories contain historical patterns that can guide new repairs. They argue that traditional "forward reasoning" and search-based methods (like MCTS) are often inefficient and prone to "reasoning drift" because they operate under outcome uncertainty.

To resolve this, ConRAD leverages historically resolved issues within the same codebase to provide guidance for new bugs. The framework follows a three-stage pipeline:

- Exemplar Mining: Identifying a semantically similar past bug using a combination of textual similarity and an LLM-based judge.
- Exemplar Guardian: A filtering mechanism that evaluates the transferability of a retrieved fix to prevent "negative transfer" from misleading analogies.
- Backward Reasoning Distillation (BRD): The system's primary innovation, which reconstructs a step-wise repair plan (targeting file localization, function localization, and patch generation) by working backward from a verified ground-truth fix.

The authors evaluate ConRAD on SWE-Bench Lite using multiple LLM backbones, including GPT-4o, DeepSeek-V3, and GPT-5. Their findings demonstrate that ConRAD significantly improves Pass@1 performance (e.g., a 10.4% increase for GPT-4o) and enhances fault localization accuracy. Furthermore, the paper highlights ConRAD’s inference efficiency, showing it achieves superior results to MCTS while requiring 9.71 times fewer LLM calls and 2.47 times fewer tokens. The framework is designed to be model-agnostic and can be integrated into existing agentic scaffolds without the need for fine-tuning.

**Compliance With Llm Reviewing Policy:**

Affirmed.

**Key Questions For Authors:**

- Have you tested the framework's performance on issues that require changes to four or more files?
- Does the reported overhead for ConRAD in Table 5 include the total token and call count for all three stages (Mining, Guardian, and Distillation), or only the final inference-time repair?
- How does the system perform in repositories with sparse history (e.g., <50 resolved issues), and is there a fallback mechanism for when the Exemplar Guardian rejects all candidates?

**Limitations:**

Yes.

**Strengths And Weaknesses:**

Strengths:

SOUNDNESS
- The paper provides extensive experimental results on SWE-Bench Lite. The claims are supported by a statistically significant improvement (McNemar’s test, p<0.001) over the baseline across three different backbones: GPT-4o, GPT-5, and DeepSeek-V3.
- The methodology is validated through multiple ablations. For example, the authors demonstrate that the Exemplar Guardian is critical (e.g., a 5.6% drop in Pass@1 without it for DeepSeek-V3).
- The authors evaluated performance across different difficulty levels (Easy vs. Medium-High) based on external annotations, demonstrating that ConRAD's benefits are not limited to trivial fixes

PRESENTATION
- The work clearly positions itself against existing agentic systems (e.g., SWE-agent, OpenHands) and search-based strategies (e.g., MCTS).
- The overall narrative is easy to follow. Specifically, the paper clearly decomposes the framework into a three-stage workflow: mining, filtering (Guardian), and distillation.
- The submission provides extensive details in the appendices, including full prompt templates for all three stages.

SIGNIFICANCE
- The paper demonstrates that backward distillation can achieve superior results (+13% Pass@1) than MCTS while using 9.71 times fewer LLM calls and 2.47 times fewer tokens. This is promising for "long-horizon" tasks.
- The paper shows that distilled plans improve not just the final patch but also fault localization accuracy (e.g., an 8.3% improvement in function localization for GPT-4o), suggesting the guidance is valuable throughout the entire pipeline.
- Since it is an inference-time augmentation that requires no fine-tuning, it can be used with various LLM backbones.

ORIGINALITY
- The introduction of Backward Reasoning Distillation (BRD)—reconstructing reasoning backward from verified outcomes—is a departure from the dominant forward-exploration methods.
- The work treats historical fixes not just as examples, but as outcome-conditioned constraints.


---------

Weaknesses:

SOUNDNESS
- The evaluation is limited to SWE-Bench Lite, which focuses on Python projects.Thus, the framework's effectiveness in other programming languages or software domains (e.g., embedded systems or low-level C/C++ code) is unverified.
- During the data construction phase, the authors restricted historical exemplars to those where the ground-truth patch modified at most three source files. This was done to ensure the reasoning remains interpretable and transferable, but it may limit the framework's ability to learn from or assist in large-scale refactorings or complex multi-file repairs.
- Both the Exemplar Guardian's filtering decisions and the distillation of reasoning plans depend on the underlying LLM's own reasoning capabilities. If the model generates unreliable judgments or flawed step-wise plans, the quality of the repair guidance will suffer.

PRESENTATION
- The "Background" section begins with the LLM era (localized patch synthesis and conversational refinement), offering very little information on traditional, non-LLM APR techniques.
- The authors state that limiting historical exemplars to those modifying at most three files ensures they are "interpretable and transferable". However, the paper does not define a metric for "interpretability" or provide evidence (such as a user study) that these plans are indeed easier for a human or a model to interpret than plans derived from more fixes.

SIGNIFICANCE
- ConRAD relies on the availability of historically resolved issues and verified fixes within the same repository. Projects with sparse or non-existent bug-fix histories would likely see reduced or no benefit from this framework.
- The significance of the "distillation" contribution is heavily gated by the initial retrieval stage. Results show that using the "Top-3" retrieved exemplar instead of the "Top-1" leads to a 6% drop in Pass@1. This sensitivity suggests that the framework's practical significance is highly dependent on the accuracy of the semantic alignment judge; if the initial retrieval fails to find the absolute best match, the value of the distilled plan diminishes significantly.
- ConRAD is explicitly designed as an "inference-time augmentation" rather than a ground-up repair system. It depends heavily on existing agentic scaffolds like Agentless or SWE-agent to perform the actual repair actions.

ORIGINALITY
- While the paper presents Backward Reasoning Distillation (BRD) as its primary innovation, it acknowledges that "retrospective reasoning reconstruction has shown promise in open-ended tasks".
- Several key components of the framework are adaptations of existing methodologies. The "LLM-as-a-judge" approach used in Stage 1 is based on prior work. Similarly, the Exemplar Guardian's structured rubric and filtering mechanism are "inspired by meta-prompting principles" and evaluation criteria from concurrent literature. The originality lies in the combination of these parts rather than the invention of the underlying mechanisms.
- The three-stage process—mining, filtering, and prompt injection—closely mirrors the architecture of advanced Retrieval-Augmented Generation (RAG) systems.

---

> ### Author Rebuttal · Authors · 2026-03-31
>
> Thank you very much for the feedback. We conducted additional experiments using a **new agent scaffold**, mini-swe-agent, and clarified other confusions.
>
>
> >### Q1/W2. Issues that require changes to multiple  files
>
> We evaluate ConRAD in the bug-fixing setting. Prior empirical studies show that most bug fixes involve changes to three or fewer files [1], and we use the same threshold as a quality-control filter when constructing exemplars. We observe similar situations in SWE Bench, where almost all the fixes happen within one or two files. A prior study [2] finds that larger commits are more likely to include unrelated changes or non-bug-fixing activities.
>
> \[1\] Ferreira, Mívian, et al. "Inside commits: An empirical study on commits in open-source software." Proceedings of the XXXV Brazilian Symposium on Software Engineering. 2021.
> \[2\] Herzig, Kim, Sascha Just, and Andreas Zeller. "The impact of tangled code changes on defect prediction models." Empirical Software Engineering 21.2 (2016): 303-336.
>
> >### Q2. Does the reported overhead for ConRAD count for all three stages or only the final inference-time repair?
>
> The overhead reported in Table 5 is the **full end-to-end** cost. It includes the LLM calls and token usage for all three ConRAD stages, Mining, Guardian, and Distillation, as well as the repair inference. We will clarify this in the revision.
>
> >### Q3/W4. ConRAD’s performance in repositories with sparse history and fallback mechanisms.
>
> When no suitable exemplar is found, the Exemplar Guardian falls back to the unmodified repair agent. Table 2 compares ConRAD with and without the Guardian; across models, adding EG improves Pass@1 by 2.3%–5.6%.
>
> We also analyze repositories with different numbers of eligible historical issues. Django (1,032 historical issues) improves by +9.4% / +5.5%, while Sympy (only 230\) still shows \+7.5%/+12.5% (GPT-4o/5). ConRAD falls back to the original scaffold when no matching issue is found, and improves performance when one is available.
>
>
>
>
> | Repo | Hist. Issues | In Lite | GPT-4o (New resolved / Impr.) | GPT-5 (New resolved / Impr.) |
> | :---: | :---: | :---: | :---: | :---: |
> | Django | 1,032 | 127 | \+12 / \+9.4% | \+7 / \+5.5% |
> | Sympy | 230 | 80 | \+6 / \+7.5% | \+10 / \+12.5% |
>
>
>
>
>
> >### W1: Evaluation only focuses on Python Language.
>
> We chose SWE-Bench Lite because it is a standard benchmark for repository-level APR. Although we evaluated on Python, **ConRAD is not tied to language-specific analysis**: its distilled plans are expressed in natural language and operate over issue descriptions, code diffs, and repair outcomes.
>
>
>
> >### W3: Repair guidance quality depends on the underlying LLM
>
> This is a shared challenge for LLM-based reasoning approaches. However, ConRAD reduces the reasoning burden compared to forward methods like MCTS: instead of generating and evaluating reasoning under outcome uncertainty, ConRAD provides the verified fix as input, so the model only needs to reconstruct an explanation for a known-correct outcome. Under the same setting, ConRAD outperforms MCTS by \+13% Pass@1 while using 9.71x fewer LLM calls.
>
> >### W5: Retrieval sensitivity: Top-1 vs Top-3 drops 6%
>
> Retrieval quality affects performance, but does not fully determine ConRAD’s value. The 6% gap between Top-1 and Top-3 reflects the benefit of better alignment, but even with the weaker Top-3 exemplar, ConRAD still improves over the baseline by \+5% Pass@1. This means that our reasoning distillation still provides benefits even when the issues are less similar.
>
> >### W6: ConRAD depends on existing agentic scaffolds like Agentless.
>
> ConRAD is designed as an inference-time framework that augments existing APR scaffolds. This design isolates reasoning distillation's effect while ensuring easy integration into existing pipelines.
>
> To verify its benefit is not scaffold-specific, we **added a new experiment and evaluated ConRAD on the mini-SWE-agent** over 100 randomly sampled issues from SWE-Bench Lite with GPT-5. ConRAD **improves the mini-SWE-agent by \+10.0%** on this subset, matching the \+10.3% gain on Agentless over the full SWE-bench lite issues, suggesting the benefit transfers across architecturally different scaffolds.
>
> | Scaffold | Pass@1 |
> |---|---:|
> | mini-swe-agent | 48.0% (48/100) |
> | mini-swe-agent + ConRAD | 58.0% (58/100) |
>
>
> >### W7: Novelty relative  to prior reasoning reconstruction work.
>
> We agree that retrospective reasoning reconstruction is conceptually similar to our approach, but **only at a very high level**. At the technical level, the two are not comparable. Repository-level APR applies substantially different requirements: the reconstructed reasoning must be grounded in code, remain coherent across three tightly coupled stages (file and function localization, and patch generation), and transfer across issues. ConRAD is therefore not a direct application or extension of prior work, but a distinct formulation for multi-stage program repair.

---

> > ### Author Rebuttal · Reviewer_EuHZ · 2026-04-03
> >
> > Thank you for the detailed rebuttal. The responses to most concerns were thorough and convincing---in particular, the empirical citations justifying the three-file threshold for transferability, the clarification that Table 5 reports full end-to-end costs, the per-repository sparse history analysis, and the new mini-SWE-agent experiment demonstrating scaffold-agnostic gains all directly addressed the raised concerns.
> >
> > However, the response to the interpretability concern remains incomplete. While the authors provide strong empirical justification for the three-file threshold on transferability grounds, the claim that this threshold also ensures "interpretability" is still unsubstantiated. The rebuttal does not define what interpretability means in this context, nor does it offer any evidence---such as a proxy measure or human evaluation---that distilled plans derived from smaller patches are meaningfully easier for a model or human to follow. Since interpretability is one of the two stated justifications for this design choice, leaving it unaddressed is a gap worth noting for the camera-ready.

---

> > > ### Author Response · Authors · 2026-04-07
> > >
> > > We appreciate the feedback. Due to the character limit in our initial rebuttal, we did not fully address the concern about the definition of interpretability. We now clarify this definition and provide supporting evidence.
> > >
> > > **Definition.** In our context, *interpretability* refers to whether a distilled plan can be readily followed as a single, self-consistent repair rationale by the downstream model. We operationalize this using three criteria:
> > >
> > > - **Low redundancy:** the distilled plan does not repeat the same diagnosis or repair point unnecessarily across reasoning steps.
> > > - **Logical continuity:** the reasoning in the distilled plan progresses from diagnosis to fix without abrupt or unsupported jumps, with clear causal links between reasoning steps.
> > > - **Followability:** the stages of the distilled plan collectively form an end-to-end repair rationale whose intermediate outcomes consistently point toward the final fix.
> > >
> > > >### Evaluation
> > >
> > > To evaluate this, we collected **50 historical issues involving more than 3 edited files** from the repositories associated with SWE-bench Lite issues, and constructed **50 comparison pairs** between distilled plans derived from these issues and distilled plans derived from exemplars with ≤3 edited files.
> > >
> > >
> > > **1. Manual evaluation:**
> > > Two authors independently evaluated the same 50 anonymized plan pairs, with presentation order randomized, using the three criteria above (**Cohen’s kappa = 0.846**). Disagreements were resolved through discussion.
> > >
> > > | Evaluation | Judged as more interpretable (≤3-file exemplar) | Judged as more interpretable (>3-file exemplar) |
> > > |---|:---:|:---:|
> > > | **Manual evaluation** | 46/50 (92%) | 4/50 (8%) |
> > >
> > > Under these criteria, the manual evaluation more often judges distilled plans derived from ≤3-file exemplars to be more interpretable.
> > >
> > > **2. LLM evaluation:** To further examine whether this preference also holds from the perspective of a downstream model, we performed an LLM-as-a-judge comparison on the same 50 pairs using the same three criteria as explicit instructions.
> > >
> > > | Evaluation | Judged as more interpretable (≤3-file exemplar) | Judged as more interpretable (>3-file exemplar) |
> > > |---|:---:|:---:|
> > > | **LLM evaluation** | 44/50 (88%) | 6/50 (12%) |
> > >
> > > The same trend largely holds in model-based evaluation: distilled plans from ≤3-file exemplars are also more often judged as more interpretable under the same rubric.
> > >
> > > **Overall consistency.** The manual and LLM-assisted views are broadly consistent:
> > >
> > > | Comparison | Value |
> > > |---|:---:|
> > > | Agreement | 40/50 (80%) |
> > > | McNemar exact test (two-sided) | p = 0.754 |
> > >
> > > This indicates no systematic disagreement between the two views.
> > >
> > > **3. Static metrics (supplementary proxy):** While compactness alone does not directly establish interpretability, the length difference further corroborates that >3-file exemplars produce longer distilled plans with more reasoning steps.
> > >
> > > | Group | n | Avg plan words | Avg steps |
> > > |---|:---:|:---:|:---:|
> > > | ≤3 files | 50 | 1,430 | 39.4 |
> > > | >3 files | 50 | 2,370 | 51.9 |
> > >
> > > Plans from >3-file exemplars are ~66% longer and contain more redundant reasoning steps.
> > >
> > > >### Final clarification
> > >
> > > We agree that this concept should be explicitly defined in the paper. The results above provide evidence for the intended sense in which we use the term: distilled plans derived from exemplars with fewer edited files are more often judged to be less redundant, more coherent, and easier to follow as end-to-end repair rationales. In the revised version, we will provide a clearer definition of this notion of interpretability and explain the evaluation protocol in more detail.
> > >
> > > Thank you again for the valuable feedback that helped strengthen our work.

---

### Decision · Program_Chairs · 2026-04-30

**Decision:**

Accept (regular)

**Comment:**

This paper proposed conditional Reasoning Distillation (ConRAD), a framework that is designed to improve the repository-level automated program repair using LLMs. Existing methods treat bugs as isolated events or rely on heavy search and also potential limitations around long context. This paper leverages existing history, finds similar issues and most importantly “The system's primary innovation, which reconstructs a step-wise repair plan (targeting file localization, function localization, and patch generation) by working backward from a verified ground-truth fix.” – quote reviewer EuHZ. This approach also boosts the performance compared to gpt-4o while saving compute from heavy MCTS.

Overall reviewers like the performance gain while at the same time being more efficient compared to MCTS. The idea of reconstructing reasoning backward from a known outcome is also interesting. While some limitations are discussed during rebuttal, I feel this paper made an interesting perspective to the field of LLM for software engineering. Definitely there are some future works we hope the authors will continue to improve, particularly in terms of the scope of even harder refactoring and multi-file repairs.